# The Effects of Various Food Products on Bisphosphonate’s Availability

**DOI:** 10.3390/pharmaceutics14040717

**Published:** 2022-03-27

**Authors:** Monika Zielińska, Grzegorz Garbacz, Jaroslaw Sczodrok, Adam Voelkel

**Affiliations:** 1Institute of Chemical Technology and Engineering, Poznań University of Technology, ul. Berdychowo 4, 60-965 Poznan, Poland; adam.voelkel@put.poznan.pl; 2Physiolution Polska sp. z o.o., Piłsudskiego 74, 50-020 Wrocław, Poland; ggarbacz@physiolution.eu; 3Physiolution GmbH, Walther-Rathenau-Straße 49 a, 17489 Greifswald, Germany; jsczodrok@physiolution.eu

**Keywords:** bisphosphonates, gastrointestinal absorption, drug bioavailability

## Abstract

The bioavailability of orally administered bisphosphonates is very low (<1%) due to their short absorption window in the proximal duodenum and high affinity for food. Food ingredients are able to bind the drug, but the presence of food extends the residence time of bisphosphonates in the absorption window. Therefore, the main goal of this study is to select a group of food products that are characterized by low binding affinity to bisphosphonates and thus will not reduce their availability upon concomitant administration. For this purpose, a combination of three methods was applied: (1) evaluation of sorption capacity for rows of digested food samples in a simulated intestinal environment; (2) evaluation of drug availability in simulated chyme; and (3) evaluation of drug availability using a simulating needle device. The results indicate that food products such as egg white and white bread are most suitable for consumption during oral bisphosphonate intake.

## 1. Introduction

Bisphosphonates are active compounds recommended for the treatment of bone diseases, assigned to BCS III class, and occur as well-soluble but poorly absorbable substances. Their low bioavailability (about 1–5% when taken orally) is due to the presence of phosphate groups that hardly penetrate the lipid layer of cell membranes. Thus, only about 50% of the absorbed drug undergoes selective retention in the skeletal system, while the rest is removed by the kidneys in an unchanged form [1].

Bisphosphonate absorption is further decreased when the drug is ingested together with food, especially food products rich in divalent cations, e.g., calcium ions, due to the formation of insoluble chelate complexes of calcium ions with bisphosphonates. Thus, a decrease in absorption efficiency is caused by the presence of food in the gastrointestinal tract [2]. Many factors have an impact on transit time, drug permeability, and systemic availability, and can cause physiological changes in the gastrointestinal tract. These factors include the meal volume, and the nutrient and caloric contents of the meal. Therefore, it is important to investigate the behavior of compounds in the intraluminal environment in both fasted and fed states [3,4].

A study on healthy volunteers was carried out to examine the food intake effects on the absorption of orally administered risedronate in order to choose the best timing regimen of its administration. It has been reported that the most beneficial impact is achieved when taking the drug on an empty stomach and continuing the fasted state, which is not recommended in clinical practice [5]. Risedronate can be detected in the blood plasma at the highest concentration as soon as 0.5 h after intake when administered on empty stomach. According to various sources, the time it takes to reach the maximum concentration varies from 4 h, when the drug is taken before a meal [6], to 2 h, when it is taken after a meal [7]. Other sources show that the best alternate regimen could be to eat breakfast at least 30 min after drug intake [5]. Bisphosphonates are absorbed in the upper part of the intestine in about one hour after ingestion [2]. However, this time depends on the composition and quantity of food ingested concomitantly with the medication.

Bisphosphonates are absorbed throughout the gastrointestinal tract, but primarily in the upper part of the small intestine, since the duodenum and jejunum have the largest surface area through which the drugs are absorbed and transferred into blood circulation. Their absorption is typically around 0.7% for the most commonly used nitrogen-containing compounds such as alendronate or risedronate, and the simultaneous consumption of foods and caloric drinks are believed to cause further reduction of absorption [8]. Taking drugs on an empty stomach is not only unrecommendable but can lead to discontinuation of treatment. However, when taken with food, the treatment may fail due to the lack of absorption [9]. Therefore, the recommendations in the leaflets of bisphosphonate medicines are as follows: take a tablet at least 30 min before the intake of the first portion of food, caloric beverages, or other medicine of the day; take the pill in an upright position (standing or sitting) to avoid heartburn and do not lie down for 30 min after taking it.

On the basis of the hitherto knowledge, the following two theses can be formulated:most food products will reduce the absorption of bisphosphonate;eating a meal in a short period after/before taking bisphosphonates will slow down the gastric emptying kinetics of API (the active pharmaceutical ingredient). It may extend the residence time of the API in the stomach and result in its longer prevalence in the absorption window in the proximal duodenum. Interestingly, if the food does not bind bisphosphonates, the increase in the systemic absorption becomes very likely.

Bisphosphonate administration concomitant with food and supplements containing divalent ions is not advisable. However, it is worth considering consuming anti-nutritional products at this time. Phytic acid has a strong chelating effect and therefore has the ability to bind to minerals. The elements supplied with food combine with phytic acid in the gastrointestinal tract and form insoluble complexes, so they cannot be absorbed into the bloodstream [10]. The influence of phosphate on the sorption of risedronate on the model food products was a collateral goal of this work.

The aim of this study was to examine a series of food products in terms of their sorption capacity of bisphosphonates. A series of commonly eaten breakfast products was selected, and a procedure based on the following three methods (Figure 1) was applied for the purpose of the study:(1)evaluation of the sorption capacity of a series of digested food samples in a simulated intestinal fluid environment;(2)evaluation of the drug availability after oral administration with a given food environment in simulated conditions;(3)evaluation of drug sorption using a simulating needle device.

## 2. Materials and Methods

### 2.1. Materials

Sodium chloride (99%), sodium hydroxide pellets (≥97.0%), acetonitrile for HPLC gradient grade, and hydrochloric acid (36–38%) were obtained from Avantor Performance Materials Poland S.A (Gliwice, Poland). Sodium phosphate monobasic (≥99.0%), pepsin, and pancreatin were obtained from Sigma-Aldrich (Steinheim, Germany). FaSSIF/FeSSIF/FaSSGF, formerly known as ‘SIF Powder’ was obtained from Biorelevant.com Ltd., (London, UK). Sodium risedronate (RSD) was obtained from Sigma-Aldrich. Stainless steel needles (ID 2.7 mm) were obtained from Danlab (Białystok, Poland). All food products were purchased from one store of the commercial network (Table 1).

### 2.2. Simulated Body Fluid

#### 2.2.1. Simulated Gastric Fluid (SGF)

The simulated gastric fluid was prepared by dissolving NaCl (2 g L^−1^) and pepsin (2 g L^−1^ equal to 2000 units/mL) in a NaHCO_3_ solution (0.42 g L^−1^). The pH of the obtained solution was adjusted to 2 using HCl. After introducing the food products, the pH was adjusted to 3 with HCl/NaOH, and then monitored continuously.

#### 2.2.2. Simulated Intestinal Fluid (SIF)

The simulated intestinal fluid was prepared by dissolving ‘SIF powder’ (2.24 g L^−1^) and NaCl (6.18 g L^−1^) in a NaHCO_3_ solution (0.42 g L^−1^). The pH of the obtained solution was adjusted to 7 using HCl/NaOH, and then monitored continuously. SIF concentrate in ten times higher concentration was also prepared.

### 2.3. Risedronate Sorption

#### 2.3.1. Sorption Capacity of Digested Food Samples

All food products were digested as follows: 5 g of food products (weighted on an analytical balance to the accuracy of ±0.1 mg) was placed in 50 mL tubes and poured with SGF (20 mL). The pH was adjusted to 3 with HCl/NaOH, and then monitored continuously for 3 h of digestion. After 3 h, the pH was adjusted to 7, and the digested food was mixed to a homogeneous consistency. It was assumed that 1 g of digestive tract content contains 0.2 g of the introduced food product. 

The digested food sorption study of risedronate was initiated by weighing 50–1000 mg of each simulated chyme sample (10–200 mg of digested food) on an analytical balance to the accuracy of ±0.1 mg and placing into polypropylene tubes (2 mL). Then, 0.2 mL of SIF concentrate, 50 µL of risedronate solution (containing 100 µg of pure risedronate), and the appropriate amount of buffer were introduced to obtain the total sample volume of 2 mL. Each sample was placed on a vortex shaker for 5 min (the sample contact time was about 10 min) and then centrifuged (15,000 rpm for 15–30 min). Next, 600 µL of each sample was transferred to a polypropylene tube, and 600 µL of methanol was added. The sorption of risedronate on each digested food product was measured with HPLC using a UV-Vis detector after centrifugation (15,000 rpm for 15 min).

In the second part, phosphate buffer was used (0.025 M) to check its effect on the sorption process efficiency. Studies were conducted for samples of 100 mg of digested food (which corresponds to 500 mg of digested tract content) with pH 7.

#### 2.3.2. Pharmaceutical Availability after Oral Administration with the Food Environment

Pharmaceutical availability evaluation was started by weighing 1 g of each food product on an analytical balance to the accuracy of ±0.1 mg and placing these samples into polypropylene tubes (15 mL). Then, 10 mL of SGF solution containing 1 mg of risedronate (0.1 mg mL^−1^) was introduced to each tube. All samples were placed onto a rotator mixer. The sorption process was carried out in such conditions for 2 h, and pH was monitored continuously. Samples were taken for HPLC tests after 10 min, 1 h, and 2 h (immediately prior simulating the intestinal conditions). A total of 300 µL was taken each time, and 600 µL methanol was added.

After two hours, 1 mL of SIF concentrate containing pancreatin (45.7 mg/mL of SIF concentrate, which is equal to 100 u/m) was introduced into each tube, and the pH was adjusted to 7. Next, the samples were taken for HPLC tests after 10 min and 1 h. A blank trial was also prepared and measured. The samples were centrifuged at 24,000 ×*g* for 15 min, and the supernatant was transferred to HPLC vials and measured using RP chromatography with UV-Vis detection.

#### 2.3.3. Drug Sorption Using a Simulating Needle Device

Conditions in the small intestine differ from those simulated in the two previous experiments. Therefore, it seemed necessary to examine the effects of selected food types during forced flow of simulated fluids containing the drug through the intestinal contents. Food products were placed into stainless steel needles, as according to the procedure described earlier [12,13]. The first stage was the introduction of the supporting layer into a needle that has a hole on the side that additionally prevents slipping out of the material. Then, the needles were filled with sorbent materials, which in this case, were the food products studied (food/sorbent mass was 100 mg).

A liquid sample containing risedronate in SIF solution (0.1 mg/mL SIF) was pumped through the in-needle extraction device to determine the sorption capacity. A piston pump LC-10A (Shimadzu Corporation, Tokyo, Japan) was used to maintain a uniform flow rate during the sorption and desorption process (0.5 mL min^−1^).

### 2.4. HPLC Analysis

The concentration of risedronate in the solutions used for the sorption tests was determined by a HP Agilent 1100 (Hewlett Packard). High-performance liquid chromatography (HPLC) was carried out on an Hypersil™ ODS C18 column (250 mm × 2.1 mm, particle size 5 μm) from Thermo Fisher Scientific (Waltham, MA, USA). The detection was performed at a wavelength of 262 nm [14,15,16]. The mobile phase was a series of mixed solutions used in the following gradients: acetonitrile (solvent A) and a 25 mM sodium phosphate monobasic pH 7 buffer (solvent B) in the range: 0–4 min 15:85 (*v*/*v*), 4–15 min 50:50 (*v*/*v*), and 15–20 min 15:85 (*v*/*v*). A calibration curve in the range of 1 to 0.002 mg mL^−1^ was prepared for risedronate (R^2^ = 0.9981).

### 2.5. Data Evaluation

Each final result presented in this paper is an average of five repeated measurements. A Dixon’s Q test was applied to determine and reject statistical outliers with a critical value of 0.71 at 95% confidence.

## 3. Results and Discussion

There are many tablets containing risedronate on the market. The most common concentration of the active substance is 30–150 mg per tablet. The 100 mg tablets were used in this study. In the experiment described in Section 3.1, the mass of digested food was increased to determine the sorption capacity. In the experiment described in Section 3.2, the time was increased to determine drug availability after oral administration within the food environment. In the experiment described in Section 3.3, the evaluation of drug sorption using a simulating needle device was performed.

### 3.1. Sorption Capacity of Digested Food Samples

The sorption capacity, expressed as the mass of the analyte that can be retained on the sorbent, in this case a digested food product, was determined for all products listed in Table 1 and results are presented in Figure 2. The consumption of products such as cheese, bran, and egg yolk was found to cause the most unfavorable impact on risedronate availability. Eating 30–40 g of these products may cause a sorption of approximately 80–90% of the dose (80–90 mg of risedronate) by this food. As indicated in Figure 2, it can be concluded that the total sorption of risedronate (100 mg) by food products occurs when 100 g of cheese or 150 g of bran or egg yolk were eaten together with the pill. Almost half of the risedronate dose (47 mg) may be sorbed by the food if 100 g of the egg white is consumed. Bearing in mind that one egg weighs about 50 g, and its white part weighs about 35 g, in this case only 11.3% of the dose will be sorbed (11.3 mg risedronate). However, consuming one whole egg will lead to the binding of more than half of the dose (mostly due to the egg yolk). One slice of toast bread (white) weighs approximately 30 g, and its consumption will lead to the binding of 30 mg of the drug. There is no significant difference between the results obtained for cheese and for bran (these products have a similar total amount of divalent ions). Comparable results were obtained for bread (whole wheat) and oat flakes (these products also have a similar total amount of divalent ions). Therefore, products containing calcium and/or magnesium will reduce the availability of bisphosphonate.

The influence of phosphate ions on the sorption of risedronate on the model food products was investigated. For this purpose, the solution of risedronate (containing 100 µg of pure risedronate), prepared as described in Section 2.3.1, was used. 

The test results given in Figure 3 indicate that, in the presence of phosphate buffer, the availability of the free fraction of bisphosphonate is increased by about 1% in the presence of bran and egg yolk, while it is increased to approximately about 10% in the presence of oatmeal, whole grain bread, and white bread. Since phosphates are well tolerated and available in a variety of food such as beverages, bread, and meat products, such interaction is favorable from the application point of view. It can be used for the development of formulations containing phosphates as well as for the adjustment of the composition of diet during pharmacotherapy to minimize the loss of the available fraction of the API upon sorption.

### 3.2. Drug Availability after Oral Administration with Food Environment

A sorption test simulating the simultaneous intake of 100 mg risedronate and 100 g of food components was conducted at a smaller scale using one gram of each food component and one milligram of risedronate. Graphs illustrating the risedronate availability over time that were obtained for the simulated digestive system during simultaneous intake of risedronate and food products are presented in Figure 4. Digestion was simulated by mixing the selected food components with fluids simulating gastric juice (SGF with pepsin) and subsequently, the small intestinal fluids with pancreatin (SIF). It can be assumed that 100 g of the white part of the egg sorbs less than 50% of the dose. As such, according to these studies, 55 mg of risedronate will still be available and absorbable by the intestine after 3 h of digestion. It can therefore be assumed that the consumption of egg white and white bread will have the lowest impact on the reduction of bioavailability of risedronate (Figure 4).

### 3.3. Drug Sorption Using Simulating Needle Device

Sorption of risedronate on non-digested foods was evaluated using a fast and simple method based on employing a special needle device. A simulating intestinal fluid was pumped through a steel needle filled with food products. Such a system forced the liquid to pass through the food/sorbent (which caused flow resistance), simulating short-term contact in the intestine. The selected products were placed individually into needles. The results from Figure 5 confirmed the previously presented results, showing that cheese absorbs the most of risedronate, while egg white absorbs the least.

## 4. Conclusions

The influence of various food products on the sorption properties of risedronate was examined using three different methods. The presented results prove that the food products with a high content of divalent ions bind high amounts of bisphosphonate. However, even those with minimal divalent ion content also reduce the bioavailability of the drug. Taking into account that it is recommended to eat a meal before taking bisphosphonates, it is essential to carefully consider which products should be taken.

The results, obtained by all three methods employed, indicate that the oral administration of risedronate should be conducted only with properly selected food components, which do not decrease the free fraction of the drug that can be absorbed. The concomitant administration of food can extend the residence time of bisphosphonates in the proximal gastrointestinal tract by causing the delay of gastric emptying. This effect can be favorable for increasing the oral bioavailability of the compounds and requires clinical investigation.

Large amounts of phosphates supplied to our body inter alia negatively affect the balance of calcium, magnesium, and zinc. During bisphosphonate administration (within a few hours), it is not advisable to eat food and supplements containing divalent ions. Because phosphorus in the diet adversely affects the absorption of divalent ions it might be beneficial to intake products that contain phosphates while taking the drug.

It has been shown that the phosphate buffer increases the availability of risedronate to varying degrees.

## Figures and Tables

**Figure 1 pharmaceutics-14-00717-f001:**
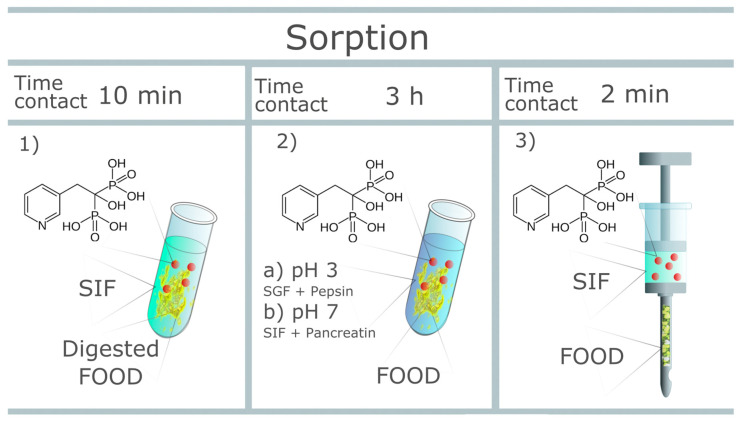
Three methods of evaluation (SIF—Simulated Intestinal Fluid; SGF—Simulated Gastric Fluid).

**Figure 2 pharmaceutics-14-00717-f002:**
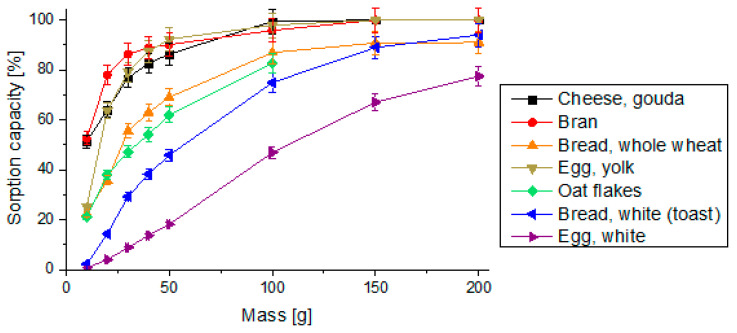
Food product sorption capacity, relative standard deviation values below 5%. The given values are expressed as the mean of five replicates (*n* = 5). SD is indicated by the error bars.

**Figure 3 pharmaceutics-14-00717-f003:**
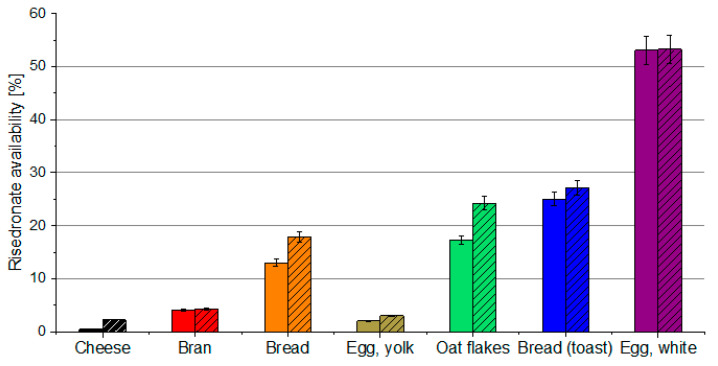
Availability of risedronate with (hatched bars) and without (solid bars) phosphate buffer. Relative standard deviation values are below 5%. The given values are expressed as the mean of five replicates (*n* = 5). SD is indicated by the error bars.

**Figure 4 pharmaceutics-14-00717-f004:**
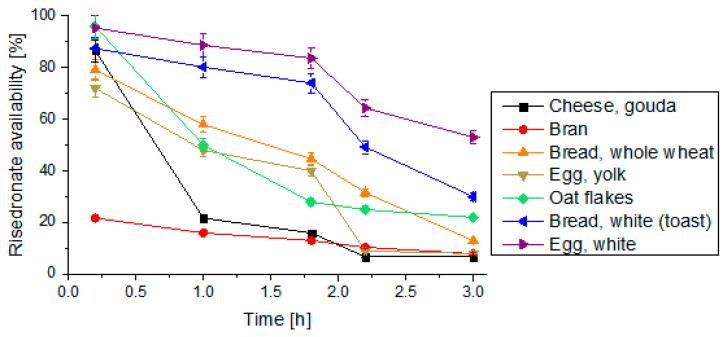
The availability of risedronate after contact with food. The tests were conducted in a simulated system using 10 mL of simulating fluids containing 1 mg of risedronate and 1 g of the food product. The relative standard deviation values are below 5%. After two hours, the simulating fluid was changed. Given values are expressed as the mean of five replicates (*n* = 5). SD is indicated by the error bars.

**Figure 5 pharmaceutics-14-00717-f005:**
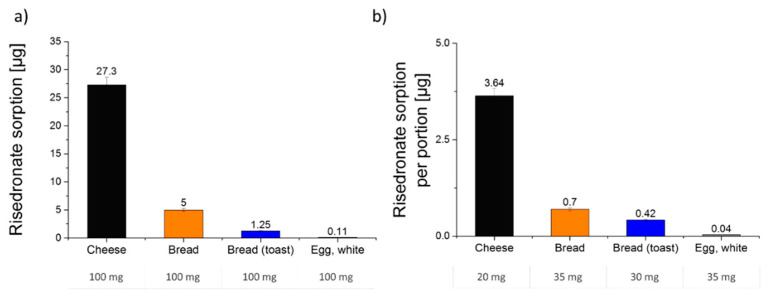
Risedronate sorption (**a**) per 100 mg of undigested food and (**b**) per single portion of food. The relative standard deviation values are below 5%. Given values are expressed as the mean of five replicates (*n* = 5). SD is indicated by the error bars.

**Table 1 pharmaceutics-14-00717-t001:** Food products and their concentration of divalent ions and fibers [11].

No.	Product	Ca^2+^ (mg/100 g)	Mg^2+^ (mg/100 g)	Fiber (mg/100 g)
1	Cheese, Gouda	700	29	0
2	Bran	389	362	29.3
3	Bread, whole wheat	161	75	6
4	Egg, yolk	129	5	0
5	Oat flakes	80	143	8
6	Bread, white (toast)	47	59	4.7
7	Egg, white	7	11	0

## Data Availability

Not applicable.

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
