# Peer review of "The Effects of Various Food Products on Bisphosphonate’s Availability"

_pharmaceutics, 2022, doi:10.3390/pharmaceutics14040717_

Round 1

Reviewer 1 Report

Bisphosphonates are known as active compounds recommended for the treatment of bone

diseases. These drugs possess low bioavailability (about 1-5% when taken orally) due to the presence of phosphate groups that hardly penetrate the lipid layer of cell membranes.

The main idea of presented paper is to select a group of food products characterized by low binding affinity to bisphosphonates, and thus these products will not reduce the availability of bisphosphonates upon their concomitant administration. The authors have used three main approaches: study of sorption capacity of digested food samples in the simulated intestinal environment; study of drug availability in simulated chyme and study of drug availability using simulating needle device. The results obtained show that egg protein and white bread caused the minimal effect on bioavailability during oral bisphosphonate intake.

So, the paper describes new and important results on rising of bisphosphonate drugs bioavailability by right selection of jointly used food products. The paper is in framework of Pharmaceutics subject and can be published in present form after some improvements of English language.

Author Response

Response to Reviewer 1 Comments

Point 1: The paper is in framework of Pharmaceutics subject and can be published in present form after some improvements of English language.

Response 1: We appreciate this opinion. We made changes to the text using the Track Changes option.

Reviewer 2 Report

In this study, authors determined the effects of various food products on bisphosphonates bioavailability by using in vitro study. Th study was performed by evaluation of sorption capacity of digested food samples in the simulated intestinal environment, simulated chyme and using simulating needle device. Overall the manuscript is written well but main limitation is the conclusion are based on the in vitro study only.

  1. The present title is not suitable. authors only described the availability of residronate and sorption capacity of different food in simulated conditions. How can you described as impact on bioavailability without performing in vivo studies.  Kindly check the defination of bioavailability and revise the conclusion of the study.
  2. Residronate belongs to class III drug, i.e low bioavailability is due to poor permeability. Then how can high availability and sorption effecinecy can enhance the permeability of drugs.
  3. In introduction, authors written that thelow bioavailability is due to the presence of phosphate groups in bisphosphonates but the results of this study report that the phosphate buffer increases the bioavailability of risedronate. Explain
  4. Authors didnot mentioned about the brand of the drug used in this study.
  5. During HPLC analysis, which concentration was measured sodium risedronate salt or residronate only. How can sodium residronate can be measure after simulation study?
  6. Residronate showed high intrasubject variability in pharmacokinetics following oral absorption so even high availaibilty of drug with these foods, how can ensure the increase in the bioavailability.

Author Response

Response to Reviewer 2 Comments

Point 1: The present title is not suitable. authors only described the availability of residronate and sorption capacity of different food in simulated conditions. How can you described as impact on bioavailability without performing in vivo studies.  Kindly check the defination of bioavailability and revise the conclusion of the study.

Response 1: The first draft had such a wrong and inappropriate title. Unfortunately, an imp broke in while overwriting files. Thank you very much for finding this error.

Point 2: Residronate belongs to class III drug, i.e low bioavailability is due to poor permeability. Then how can high availability and sorption effecinecy can enhance the permeability of drugs.

Response 2: We have been misunderstood, or, rather, we have put our thoughts into words incorrectly.

If someone will take the drug while eating food products containing large amounts of divalent ions, then the entire dose of the drug could be wasted.

Since "You cannot get water out of a stone", it will be beneficial to offer such food products that will allow the greatest possible availability of the drug, which will result in the possibility of the drug being absorbed by the intestine (the contact time will be longer).

Point 3: In introduction, authors written that the low bioavailability is due to the presence of phosphate groups in bisphosphonates but the results of this study report that the phosphate buffer increases the bioavailability of risedronate. Explain

Response 3: We adjusted the text according to the Reviewer instruction.

Introduction:

“Bisphosphonate concomitant administration with food and supplements containing di-valent ions is not advisable. However, it is worth considering consuming anti-nutritional products at this time. Phytic acid has a strong chelating effect and therefore has the ability to bind to minerals. The elements supplied with food combine with phytic acid in the gastrointestinal tract and form insoluble complexes, so they cannot be absorbed into the bloodstream [9]. The influence of phosphate on the sorption of risedronate on the model food products was a collateral goal of this work.”

Conclusions:

“Large amounts of phosphates supplied to our body inter alia negatively affect the balance of calcium, magnesium and zinc. During bisphosphonate administration (within a few hours), it is not advisable to eat food and supplements containing divalent ions. Because phosphorus in the diet adversely affects the absorption of divalent ions, therefore, it might be beneficial to intake products that contain phosphates while taking the drug.”

Point 4: Authors didnot mentioned about the brand of the drug used in this study.

Response 4: We adjusted the text according to the Reviewer instruction. We added sodium risedronate in the materials.

Point 5: During HPLC analysis, which concentration was measured sodium risedronate salt or residronate only. How can sodium residronate can be measure after simulation study?

Response 5: We adjusted the text according to the Reviewer instruction. We replaced sodium risedronate with risedronate in point 2.4.

Point 6: Residronate showed high intrasubject variability in pharmacokinetics following oral absorption so even high availaibilty of drug with these foods, how can ensure the increase in the bioavailability.

Response 6: In published studies involving volunteers, it was clearly shown that when and whether they ate a meal, it had a huge impact on bioavailability. However, the effect of the individual components of the meal has not been investigated - only records such as “breakfast” can be found. For this reason, the idea of creating this work was born.

Reviewer 3 Report

This paper describes the effects of various food products on bisphosphonates bioa-2 vailability.   The author has investigated that the availability of risedronate after contact with food and/or with and without phosphate buffer.  However, I point out the issues to be addressed.

Major

i). In introduction part, you need to quote a new information.  I suggest two scientific paper and a guideline regarding the intestinal motility on human and dogs.

Stappaerts J, Wuyts B, Tack J, Annaert P, Augustijns P. Human and simulated intestinal fluids as solvent systems to explore food effects on intestinal solubility and permeability. Eur J Pharm Sci 2014; 63:178-186.

Miyake M, Oka Y, Mukai T. Food effect on meal administration time of pharmacokinetic profile of cilostazol, a BCS class II drug. Xenobiotica. 2020; 50:145-149.

Food and Drug Administration (FDA). Guidance for Industry: Food-Effect Bioavailability and Fed Bioequivalence Studies. Rockville, MD: US Department of Health and Human Services, Food and Drug Administration, Center for Drug Evaluation and Research. 2002.

  1. ii) In page 4, you describe about simulated fluids on gastric (SGF) and intestinal (SIF). Is it your original or you used USP buffer? You need to write more information about them because lots of simulated fluid has been reported already.

Minor

  1. i) Line 109, you need space before g. Line 121 is, too.
  2. ii) In references, you should space correctly, on reference 11 and 12.

Author Response

Response to Reviewer 3 Comments

Point 1: In introduction part, you need to quote a new information.  I suggest two scientific paper and a guideline regarding the intestinal motility on human and dogs.

Stappaerts J, Wuyts B, Tack J, Annaert P, Augustijns P. Human and simulated intestinal fluids as solvent systems to explore food effects on intestinal solubility and permeability. Eur J Pharm Sci 2014; 63:178-186.

Miyake M, Oka Y, Mukai T. Food effect on meal administration time of pharmacokinetic profile of cilostazol, a BCS class II drug. Xenobiotica. 2020; 50:145-149.

Food and Drug Administration (FDA). Guidance for Industry: Food-Effect Bioavailability and Fed Bioequivalence Studies. Rockville, MD: US Department of Health and Human Services, Food and Drug Administration, Center for Drug Evaluation and Research. 2002.

Response 1: We extended the introduction and we added a reference.

Point 2: In page 4, you describe about simulated fluids on gastric (SGF) and intestinal (SIF). Is it your original or you used USP buffer? You need to write more information about them because lots of simulated fluid has been reported already.

Response 2: We described the preparation methods in points 2.2.1 and 2.2.2.

FaSSIF/ FeSSIF/ FaSSGF, formerly known as 'SIF Powder' were obtained from Biorelevant.com Ltd (London, United Kingdom)

Point 3: Line 109, you need space before g. Line 121 is, too.

Response 3: We adjusted the text according to the Reviewer instruction.

Point 4: In references, you should space correctly, on reference 11 and 12.

Response 4: We adjusted the text according to the Reviewer instruction.  

Reviewer 4 Report

This study (pharmaceutics-1622500) was conducted to examine a series of food products in terms of their sorption capacity of bisphosphonates using commonly eaten breakfast products. Interesting data are obtained; however, some additional comments are needed, I think.

Please check the following comments:

Major

  • Please comment about the relationship between the sorption of risedronate (Fig. 2) and contents of metal ions (Ca, Mg, and/or Ca/Mg in Table 1) in food products.
  • Please add some comments regarding the mechanism why PBS or phosphate-containing beverages can decrease the sorption of bisphosphonates (Fig. 3).

Minor

  • Line 43-44: It is written that “Bisphosphonates are absorbed in the upper part of the intestine in about one hour after ingestion” Please explain the mechanism about this briefly.
  • Page 3, Fig. 1. The sorption of risedronate was examined by using three different methods with different incubation times. How did you determine each incubation time?
  • Line 254: It is written that “The use of phosphate buffer was considered during the studies” Is this sentence OK?

Author Response

Response to Reviewer 4 Comments

Point 1: Please comment about the relationship between the sorption of risedronate (Fig. 2) and contents of metal ions (Ca, Mg, and/or Ca/Mg in Table 1) in food products.

Response 1: We adjusted the text according to the Reviewer instruction.

“There is no significant difference between the results obtained for cheese and for bran (There is no significant difference between the results obtained for cheese and for bran (these products have a similar total amount of divalent ions). Comparable results were obtained for bread (whole wheat) and oat flakes as well (similar total amount of divalent ions). Therefore, products containing calcium and / or magnesium will reduce the availability of the bisphosphonate.”

Point 2: Please add some comments regarding the mechanism why PBS or phosphate-containing beverages can decrease the sorption of bisphosphonates (Fig. 3).

Response 2: We adjusted the text according to the Reviewer instruction.

“Large amounts of phosphates supplied to our body inter alia negatively affect the balance of calcium, magnesium and zinc. During bisphosphonate administration (within a few hours), it is not advisable to eat food and supplements containing divalent ions. Because phosphorus in the diet adversely affects the absorption of divalent ions, therefore, it might be beneficial to intake products that contain phosphates while taking the drug.”

Point 3: Line 43-44: It is written that “Bisphosphonates are absorbed in the upper part of the intestine in about one hour after ingestion” Please explain the mechanism about this briefly.

Response 3: In the next paragraph we wrote: “Bisphosphonates are absorbed throughout the gastrointestinal tract, but primarily in the upper part of the small intestine since the duodenum and jejunum have the largest surface area through which the drugs are absorbed and transferred into blood circulation.”

If the question is about "one hour" then, this conclusion comes from book:

[2] Bartl R, Frisch B, Tresckow B, Bartl C, Bisphosphonates in medical practice: Actions-Side effects-Indications- Strategies, Springer-Verlag Berlin Heidelberg, 2007.

Point 4: Page 3, Fig. 1. The sorption of risedronate was examined by using three different methods with different incubation times. How did you determine each incubation time?

Response 4: Three methods was applied:

  • evaluation of sorption capacity of a series of digested food samples in the simulated intestinal fluid environment;

10 minutes was determined on the basis of preliminary research

  • evaluation of drug availability after oral administration with a given food envi-ronment in simulated conditions;

3 hours was chosen due to the mean residence time of the digested tract content

  • evaluation of drug sorption using a simulating needle device.

Based on previous research it was determined that a flow rate of 0.5 ml per minute is preferable.

Point 5: Line 254: It is written that “The use of phosphate buffer was considered during the studies” Is this sentence OK?

Response 5: This sentence has been deleted.

Round 2

Reviewer 2 Report

Authors addressed the comments carefully and now can be consider for publication

Reviewer 3 Report

The author has improved your manuscript.  However, I point out the issues to be addressed.

Major

i). In introduction and discussion part, you need to write the detail of food effect and intestinal motility more.  Because, you need to describe the animal motility and food effect more included human.  Accordingly, I suggest a scientific paper and a guideline regarding the intestinal motility on human and dogs.

Miyake M, Oka Y, Mukai T. Food effect on meal administration time of pharmacokinetic profile of cilostazol, a BCS class II drug. Xenobiotica. 2020; 50:145-149.

Food and Drug Administration (FDA). Guidance for Industry: Food-Effect Bioavailability and Fed Bioequivalence Studies. Rockville, MD: US Department of Health and Human Services, Food and Drug Administration, Center for Drug Evaluation and Research. 2002.

Round 3

Reviewer 3 Report

I can accept.